# Factors associated with uptake of human papilloma virus vaccine among adolescent girls: A cross sectional survey on insights into HPV Infection Prevention in Kabarole District, Western Uganda

**Solomon Asiimwe[1], Fred N. Bagenda[1], Tony Mugisa[2]***

1 Department of Community Health, Mbarara University of Science and Technology, Mbarara, Uganda,
2 Department of Public Health, Faculty of Health Sciences, Mountains of the Moon University, Fort Portal, Uganda

* tonyamugisa2013@gmail.com

## Abstract

### Background

Human papilloma virus (HPV) infection imposes a substantial global disease burden and represents a critical public health concern. The persistently low uptake of HPV vaccination poses a significant obstacle to reducing cervical cancer incidence, particularly in remote rural areas of developing nations.

### Objective

This study aims to assess the extent of Human papilloma virus vaccine coverage among adolescents and explore the associated determinants to facilitate enhanced planning strategies within the Kabarole district.

### Methodology

Employing a cross-sectional survey approach, data were gathered from 240 adolescent girls residing in the Kabarole district between 01/09/2022 and 18/10/2022. Simple and multiple logistic regression analyses were employed to investigate the relationships between HPV vaccination uptake and various independent variables, including demographics, HPV knowledge, and health system factors.

### Results

Of the 240 adolescents enrolled, the overall prevalence of uptake of HPV vaccination was 63%. The uptake of human papilloma was associated with demographics knowledge about HPV and health systems factors. The multivariate analysis showed that parents who completed secondary level of education were 4.1 times more likely to take their children for HPV vaccination compared to parents whose education level was primary or had no

**Data availability statement:** All relevant data are within the paper and its Supporting Information files.

**Funding:** The author(s) received no specific funding for this work.

**Competing interests:** The Authors have declared that no competing interests exist.

**Abbreviations:** Cacx: Cancer of the cervix; CI: Confidence Interval; ELISA: Enzyme Linked Immune Absorbent Assays; FREC: Faculty Research and Ethics Committee; HCIII: Health Center Three; HIV: Human Immune Virus; HMIS: Health Management Information Systems; HPV: Human papilloma virus; HPV 1: Human papilloma virus vaccine first dose; HPV 2: Human papilloma virus vaccine second dose; MoH: Ministry of Health; MUST: Mbarara University of Science and Technology; PCR: Polymerase Chain Reaction; RDTs: Rapid Diagnostic Test; REC: Research and Ethics Committee; UDHS: Uganda Demographic Health Surveys; UBOS: Uganda Bureau of Statistics; UGX: Uganda Shillings; VHT: Village Health Team; WHO: World Health Organization.

formal education (AOR = 4.06; 95%CI (1.69 - 6.87); p = 0.004). Distance from home to facility was associated with uptake HPV vaccination. Participants who came from the distance of more than 5 km were 60% less likely to uptake HPV vaccination compared to those who come from 5km or less (OR = 0.4; 95%CI (0.34 − 0.89); p = 0.006). Results further revealed that parents whose knowledge about HPV vaccination was moderate were three times more likely to take up HPV vaccine compared to those whose knowledge was low (OR = 2.99; 95%CI(1.14 − 7.87); p = 0.026).

## Conclusion

HPV vaccination uptake was at 63% and relatively lower than national average. Education of parents, Knowledge of HPV vaccination and distance to facility were factors significantly associated with uptake of human papilloma virus vaccine.

## Introduction

Human papilloma virus (HPV) infection has created a significant disease burden worldwide and is an important topic in public health [1]. HPV infection was the most common sexually transmitted infection [2]. It was estimated that 75% of sexually active people were infected with HPV during their lifetime (Weaver et al., 2022). There are many genotypes of the HPV [3]. HPV types 6 and 11 are the cause of 90% of cases of genital warts, whereas HPV types 16 and 18 were considered to be high-risk viruses, contributing to 70% of cases of cervical cancer [4,5]. These virus sub-types underwent cytopathology changes, causing cervical intraepithelial neoplasia, which would eventually evolve to cervical cancer after approximately 2 decades [5]. Cervical cancer is the fourth leading cause of female cancer and ranks as the second most common form of cancer globally among females aged 15 to 44 years [5]. This is in comparison with other forms of common cancers including hepatobiliary, gall bladder and bile cancers [6] A 33.6% prevalence of human papilloma virus (HPV) among women in Uganda combined with low screening uptake [7] had resulted in the country having one of the highest cervical cancer incidence rates in the world [8]. This is attributed to a number of factors including early marriages, multiple sexual partners, multi-parity, sexually transmitted infections including HIV infection [9], tobacco use, vitamin deficiency and HPV infection [10].

Over 80% of diagnosed cases present with an advanced stage of the disease [11]. Cervical cancer accounted for 5.2% of the cancer burden worldwide, leading to 530 000 new cases and 270 000 deaths every year for the past decade [7]. Apart from cervical cancer, HPV can cause precancerous lesions, ano-genital warts and other cancers of the vulva, vagina, penis, anus, and oro-pharynx [12]. In Uganda, cervical cancer contributes 50–60% of all the female cancers and it is associated with a relative survival of approximately 20% [13].

A prophylactic HPV vaccine was approved and licensed in 2006 [14], and was available since then, to prevent HPV associated infections targeting females aged 9 to 26 years [15]. In late 2009, it was recommended that the quadrivalent HPV vaccine for males also be approved [16]. By 2014, the Food and Drug Administration (FDA) approved another new HPV vaccine to provide additional protection against more types of HPV [17]. Studies indicate that with an optimal coverage of about 70%, the life time risk of cervical cancer would be reduced by more than 50% [18].

Many countries now include the HPV vaccination in their national vaccination program [19]. As many as 19 countries in Europe (Austria, Belgium, Denmark, France, Germany, Greece, Iceland, Ireland, Italy, Latvia, Luxembourg, the Netherlands, Norway, Portugal,

Romania, Slovenia, Spain, Sweden, and the United Kingdom) introduced a program of routine HPV vaccinations in 2012. Coverage rates ranged from 17% to 84%, and 10 out of the 19 countries organized catch-up programs by May 2012 [20]. In Africa, a total of 21 developing countries including Uganda have implemented HPV vaccination projects among young girls under the support of a public-private partnership, the Global Alliance for Vaccines and Immunizations since 2013. An estimated 206 000 girls from low-income countries were expected to benefit from these projects [21]. Specifically in Uganda, the vaccine was introduced in November 2015 and rolled out to all the districts including Kabarole, and it was given to girls aged 10 years, two doses with a time interval of six months after the first dose [17,22]. For this study, uptake referred to completion of the two recommended doses on schedule.

## Definition of subject matter

The subject matter of the article titled "Factors Associated with Uptake of Human Papilloma virus Vaccine among Adolescent girls in Kabarole District: Insights into HPV Infection Prevention" centers around the factors influencing the acceptance and administration of the Human Papillomavirus (HPV) vaccine among adolescent girls in the Kabarole District.

The title indicates a focus on understanding the determinants affecting the adoption of the HPV vaccine among adolescent girls specifically in the Kabarole District.

The objective of the study is to explore and analyze the factors associated with the uptake of the HPV vaccine among adolescent girls in Kabarole District. This involves identifying various elements that influence the decision-making process regarding HPV vaccination.

The study findings provide insights into the prevention of HPV infection through vaccination. By examining the factors influencing vaccine uptake, the study contributes to understanding how to improve vaccination rates and ultimately reduce the prevalence of HPV infection and related diseases such as cervical cancer.

The study aims to shed light on the determinants of HPV vaccine uptake among adolescent girls in a specific geographic area, offering valuable insights for public health policymakers, healthcare providers, and community stakeholders involved in HPV infection prevention efforts.

## Significance of the study

The study findings were important for Kabarole district by providing insights on the factors that has led to the district failure to attain national targets for HPV vaccination. The district attainment of the national HPV vaccination targets is believed to reduce incidence of cervical cancer and other Human papilloma virus diseases among women. This will assist the district to join the rest of the world to strive for the achievement of the Sustainable Development Goal 3by reducing maternal mortality and preventable death among women.

The study findings also provided insights on prevalence and factors leading to the low uptake of the second HPV vaccination in the district. This information is necessary to inform policy makers, health workers and community leaders to understand the causative factors for the low uptake of the HPV vaccine so that they can develop measures to increase the uptake of this very important vaccine.

By identifying the associated factors to the low uptake of HPV vaccine, recommendations have been developed to support the district and health facilities within the district to implement blended strategies that can lead to improvement in the uptake of the HPV vaccine.

Other researchers will use information as baseline to develop proposals for studies in this area and find out more about the HPV vaccination. The study findings can also improve ministry of health planning to address the HPV vaccine uptake problem and hence reduce the prevalence of cervical cancer in the county.

## Study objectives

The study aim was to determine the uptake of human papilloma virus vaccine and associated factors among adolescent girls in Kabarole district.

The study achieved its aim using the following objectives:

1. To determine the uptake of Human papilloma virus vaccine coverage among adolescent girls aged 11 to 15 years in Kabarole district.

2. To establish the socio-demographic and socio-economic factors associated with the uptake of Human papilloma virus vaccine among adolescent girls aged 11 to 15 years in Kabarole district.

3. To establish the health system factors associated with the uptake of Human Papilloma virus vaccine among adolescent girls aged 11 to 15years in Kabarole district.

by achieving the above objectives the study aimed to answer the following questions:

1) What was the level of uptake of Human papilloma virus vaccine among adolescent girls aged 11to 15 years in Kabarole District?

2) What are socio-demographic and socio economic factors associated with the uptake of the Human papilloma virus vaccine among adolescent girls aged 11 to 15 years in Kabarole District?

3) What are the health system factors associated with the uptake of Human Papilloma virus vaccine among adolescent girls aged 11 to 15years in Kabarole district.

# Materials and methods

## Study design, setting and procedures

Between December 18/10/2021 to 18/10/2022 we conducted a study in western Uganda. A cross-sectional study design employing both quantitative and qualitative research methods was used. Actual data collection was done between 01/09/2022 and 18/10/2022. The study was done in Kabarole District, which is was one of the districts in western Uganda where HPV vaccination programme was first rolled out in the country Uganda in 2015. The district is located in Western region of Uganda, about 300 km from the city center of Kampala. The district is bordered by Bunyangabu district in the West, Kamwenge in the South, Kyenjojo in the East, Bundibugyo and Ntoroko in the North and by Kibaale in the Northeast.

The district had 38 health facilities. In terms of level of facility, the district has 4 hospitals, 2 health sub districts, 18 health center IIIs and 14 Health center IIs and all facilities conduct HPV vaccination at both static and outreach sites. The district is made up of two counties that is Burahya county and Fort portal Tourism city. The district is further subdivided into 11sub counties, 4 town councils and 3 divisions (total of 18). According to UBOS report on population survey (2017), Kabarole district has an estimated total population of 302,923 people.

The target population was adolescent female girls aged between 11 to 15 years, because they are in the age bracket of those who should have completed the Human papilloma virus vaccination schedule by the time of the study since the national roll-out of the vaccination programme in 2015. For each adolescent girl, the caretaker was interviewed to get the caretaker related factors.

The immunization focal person or in charge of the corresponding health facility in the selected sub county was engaged in a key informant interview.

The study involved multistage sampling. This method was considered appropriate because the population was large and widely scattered in the district. The primary sampling unit (PSU) was the district considered as the first sampling stage. The secondary sampling unit (SSU) were the sub-counties as the second sampling stage, parishes as the third sampling stage, households as the fourth sampling stage and then further sampling of individuals within each household selected as the fifth secondary sampling unit. From each sub county, one parish was selected by simple random sampling, where all the parishes were given numbers. The villages in each parish were be selected randomly using simple random sampling from the list of villages registered at the sub county. A total of eight [20] villages were selected. A proportion to size was used to select households and then systemic random sampling method was used. A list of households in the selected villages having adolescent girls aged between 11 and 15 years was generated using information from the village health teams. The design effect of 1.106 was factored in computation of the sample size. The sample size adjusted for design effect was 240 young girls.

For key informant interviews, 6 EPI focal persons or the in charge of a health facility which was in the selected sub county was also interviewed using a prepared checklist.

The Sample size was determined using Kish & Leslie formula stated below:

$$n = \frac{z^2}{d^2} p(1-p)$$

Where: z = the standard deviation at 95%confidence interval corresponding to 1.96, p = HPV vaccine uptake in Uganda estimated at 17% (MoH, 2016). d = Degree of error allowed set at 5% equivalent to 0.05 and n is the desired sample size.

$n = \frac{1.96^2}{0.05^2} * 0.17(1-0.17) = 217$ . Giving a calculated sample size of 217 and to cover for missing responses, and some respondents within the selected households who were feared to refuse to give consent or withdraw their consent. Therefore, the above sample size was adjusted for these factors. A 10% of potential participants were expected to withdraw their consent. The adjustment was done by dividing by 1–0.1, (217/1–0.1) = 240 (Kirkwood, 1981). A total response rate of 99.6%. A sample size of 240/241) was obtained. The study plan was to estimate the proportion of successes (vaccinated or not vaccinated), thus in a binary/dichotomous outcome variable in a single population, the appropriate formula for determining sample size was:

$$n = \frac{z^2}{d^2} p(1-p)$$

After factoring in design effect, the formula becomes:

$$n = \frac{z^2}{d^2} p(1-p) * \text{deff}$$

(**deff** - **design effect**)

Estimated sampling variance due to simple random sampling was given by

$$S^2{}_{srs} = p(1-P)/n = (0.17 * 0.83)/217 = 0.00065$$

From the data, the variance of the estimated proportion was computed as 0.0007189 $S^2{}_{correct}$ = 0.0007189. The *design effect* was then computed as:

$$Deff = \frac{S^2{}_{correct}}{S^2{}_{srs}} = \frac{0.0007189}{0.00065} = 1.106$$

Therefore, n = $1.96^2*0.17*0.83/0.05^2 *1.106 = 239.8028$
Therefore, the sample size adjusted for design effect used was = 240.

## Measurement of Knowledge about HPV vaccination

Knowledge about HPV vaccination by both the adolescent girl and the caretaker was measured using the ten standard questions in section 2 of the respective tools. Those who answered correctly any nine of the questions were considered to have high knowledge, those who answered any five questions were considered to have moderate knowledge and those who answered any four or less questions were considered to have low knowledge. These questions in the tools were tested and had a content validity index of 0.76 from two experts.

## Data Collection

Quantitative data was collected by interviews using an interviewer administered semi structured questionnaire. The data included the social demographic factors, knowledge about HPV vaccination and uptake. The adolescent girls were asked whether they have received the HPV vaccine and the number of doses with evidence and those without evidence were considered not vaccinated. The caretaker of the adolescent girl was interviewed using a separate interview guide. A content validity index of 0.76 was obtained from two experts. The questionnaires were pretested on 10 adolescent girls in a nearby village before the final use for the study. This number was selected to achieve validity and reliability, in consideration for the diversity of respondents, number of questions in the tool, and the nature of the questions. A Cronbach's alpha of 0.79 was obtained. Qualitative data were collected from key informants using a key informant's guide and the audios recorded. Data was collected between October 2021 and February 2022 [23].

## Measurement of key study variables

**Dependent variables.** The dependent variable for this study were uptake of Human papilloma virus vaccine (HPV) which was classified into fully vaccinated and not fully vaccinated, where fully vaccination means having taken all the two doses on time or schedules as evidenced by the child vaccination card or a recall of receiving two doses of the vaccine on the left upper arm.

**Independent variables .** These included individual factors such as knowledge and perception of the adolescent girls and parents/caretakers, socio-demographics, cultural factors, and health facility factors, such as availability of the HPV vaccination services in the health facilities, mode of service delivery like health facility based or outreach based, consistency of the vaccination services at the health facilities, availability of the required equipment's for HPV vaccination at the health facilities.

## Data Management and Statistical Analysis

Data collected from the questionnaires were inspected for errors and gaps. After inspection and editing, it was entered into excel version 12. Questions were coded and analyzed using STATA version 15.

Data were entered and verified in Ms-Excel and exported to. Stata 15 software. Stata Corporation, College Station, USA was used for analyses.

The data was analyzed at three different levels; univariate, bivariate and multivariate using the binary logistic regression model.

Bivariate associations were done to test any possible associations between each of the independent variables and the dependent variable. Statistical significance of the relationships was

determined for the P-value (P < 0.2) and all significant variables at this level were considered at multivariate level of analysis.

Multivariate analysis was performed to assess which factor was associated with the uptake of HPV vaccination more than the other. The HPV was classified into fully vaccinated and not fully vaccinated. Thus HPV vaccination is a nominal (Binary/dichotomous) variable and therefore the suitable model to analyze this kind of criterion variable is the binary logistic regression. Binary logistic regression analysis was used because it attempts to control for possible confounding effect of independent variables on each other and thus finds the independent association for each predictor variable with the criterion variable [11].

For qualitative data, audio tape recordings were transcribed verbatim, coded and uploaded in the qualitative data analysis software MAXQDA version 12 for analysis. Cut and paste approach was used for best quotes to triangulate quantitative information as best quotes.

## Ethics

A research proposal was first revised by the Research and ethics committee of the Faculty of Medicine of Mbarara University of science and technology. The study proposal was then submitted to Uganda National Council for Science and Technology and then approved by Mbarara University of science and technology Research and Ethics committee (MUST-REC) on 16/10/2021 under REC decision Number MUST-2021-73.

An authorization letter to carry out the study from the area was obtained from the health sub district authorities, District education officer, and the District health office. An authorization to interview health workers was obtained from the in charges of the Health facilities before participating in the study. The objectives of the study were explained to study participants or their guardians and written consent was obtained when they signed a consent form. The consent form was signed in duplicate, and one copy remained with the participant. An eligible participant was interviewed from a selected private place at his/her home to ensure privacy. Participants were explained to, all the potential discomforts that can arise out of the interview process. All participants in the study were assured that they were free to pull out of the study in case they feel uncomfortable and that no consequences at all were to be gotten for the respondents who decide to move out of the study. The respondents were told that no names were required as the information given was solely for academic research purposes.

## Results

of the 240 caretakers of female adolescents aged 11 to 15 years in Kabarole district enrolled into the study, the majority were Batoro, 153 (65.7%), married/cohabiting, 140 (52.6%), residing in a rural setting, 152 (66.3%) and 229 (95.4%) earned a net income less than 100,000 Uganda shillings. The mean age for parents/caregivers was 38.53 (SD = 8.75) years (Table 1).

HPV vaccination coverage was at 62.5% (n=150),95% CI (0.562–0.684) (Fig 1). Multivariate analysis was performed to assess which factors were associated with uptake of human papilloma virus vaccine. Binary logistic regression model was used since the outcome variable was binary/dichotomous. At a multivariate level, all factors which had p-values below the threshold of 0.2 at the bivariate analysis were included in the multivariate model (Table 2). A reference category was selected for each categorical variable.

The multivariate analysis showed that parents who completed secondary level of education were 4.1 times more likely to take their children for HPV vaccination compared to parents whose education level was primary or had no formal education Adjusted Odds Ratio (AOR = 4.06; 95%CI (1.69 - 6.87); p = 0.004). Furthermore, distance from home to facility where the services of HPV vaccination were, was also associated with uptake HPV

**Table 1. Socio-demographic characteristics of the respondents (N = 240).**

| | | n (%) |
|---|---|---|
| **Age (years)** | | |
| Adolescent | | |
| Mean = 12.71, SD = 1.33) | <13 | 107 (44.6) |
| | 13–15 | 133 (55.4) |
| Age (years) caretaker Mean = 38.53, SD = 8.75) | 25–34 years | 79 (32.9) |
| | 35–44 years | 117 (48.8) |
| | ≥44 years | 44 (18.3) |
| **Education** | ≤Primary | 215 (90.3) |
| | Secondary | 23 (23.7) |
| **Gender of the caretaker** | Male | 109 (45.4) |
| | Female | 131 (54.6) |
| **Education Adolescent** | ≤Primary | 136 (57.1) |
| | Secondary | 102 (42.9) |
| **Tribe of Adolescent** | Mutoro | 153 (65.7) |
| | Others | 87 (36.3) |
| **Religious affiliation of caretaker** | Anglican | 82 (34.3) |
| | Catholic | 114 (47.7) |
| | Others | 44 (18.3) |
| **Marital status of caretaker** | Single | 68 (28.3) |
| | Married/cohabiting | 130 (54.2) |
| | Cohabiting | 18 (7.5) |
| | Widowed/ | 6 (2.5) |
| | Divorced/separated | 18 (7.5) |
| **Residence** | Rural | 152 (66.3) |
| | Urban | 88 (36.7) |
| **Occupation of caretaker** | Unemployed | 110 (46.0) |
| | Peasant/Farmer | 108 (45.2) |
| | Business | 22 (9.2) |
| **Average monthly income (Ugandan shillings) of caretaker** | <100,000 | 229 (95.4) |
| | ≥100,000 | 11 (4.6) |

vaccination. Participants who came from the distance of more than 5 km were 60% less likely to uptake HPV vaccination compared to those who come from 5km or less (AOR = 0.4; 95%CI (0.34 – 0.89); p = 0.006). The multivariate analysis further revealed that parents whose knowledge about HPV vaccination was moderate were three times more likely to take up HPV vaccine compared to those whose knowledge was low (AOR = 2.99; 95%CI (1.14 – 7.87); p = 0.026).

Other factors like tribe of caregiver, age of parent, impact of Religion on child HPV vaccination, and impact of clan on child HPV were not significantly associated with uptake of human papilloma virus vaccine at multivariate analysis.

## Discussion

This household-based cross-sectional study among adolescent females aged 11 to 15 years in Kabarole district, HPV vaccination coverage was at 63%. Uganda Ministry of Health reported an annualized HPV coverage of 85% for HPV 1 and as low as 41% [12] for HPV 2 as of December 2017(WHO 2019). In Kabarole district, they reported high HPV1 vaccine coverage

# Factors Associated with Uptake of Human Papilloma virus Vaccine among Adolescent girls : A cross sectional survey on insights into HPV Infection Prevention in Kabarole District

## Figures in the Manuscript

**Fig 1:** *Prevalence of Human papilloma vaccination*

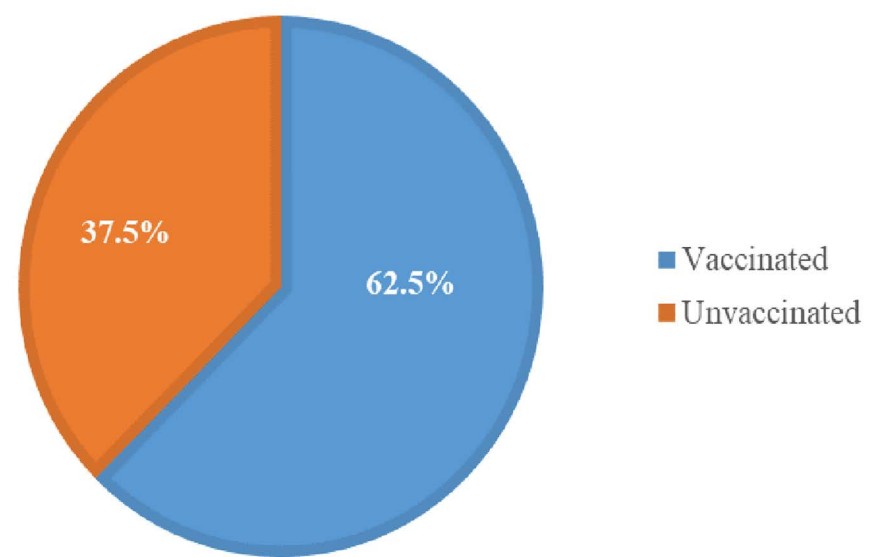

**Fig 1. Prevalence of Human papilloma vaccination.**

at 114%, but still very low HPV2 vaccine coverage at 51% in 2017 (Ministry of Health 2017), HPV1 at 110% and HPV2 at 50% in 2018 (Ministry of Health 2018) and HPV 1 at 91% and HPV2 at 64% in 2019 (Ministry of Health 2019) this is in line with the findings from the study by Chuang et al who concluded that efforts to improve HPV initiation and completion could benefit from additional attention to factors at the health care [23]. According to a study carried out in Soweto South Africa, of 224 adolescents recruited, 201 initiated the vaccine; 192 (95.5%) received a second immunization; and 164 (81.6%) completed three doses. In that qualitative study, of 39 adolescent-caregivers, it was found out that factors driving vaccine uptake reflected a socio-cultural backdrop of high HIV endemnicity, sexual violence, poverty, and an abundance of female-headed households [24,25]

Findings from a study by Migoneetal, HPV vaccine uptake in Ireland was high. Even in schools that are disadvantaged, HPV was above the national target of 80%. Since then, anti-HPV vaccine publicity has had a negative impact on national HPV vaccine uptake in Irish schools. The study also found out that even notwithstanding recent anti- HPV vaccine publicity, school-based programmes, such as the Irish HPV vaccination programme have been shown to maximize uptake of vaccines when compared with opportunistic community based programmes [26].

**Table 2. Multivariate analysis results of factors associated with the uptake of Human papilloma virus vaccine among adolescent girls aged 11 to 15 years in Kabarole district (n = 240).**

| Variables | AOR 95%CI | P_value |
|---|---|---|
| Education of caretaker | | |
| Primary/no formal education | 1 | |
| Post-Secondary | 4.06 (1.69–6.87) | **0.004**** |
| Tribe of caretaker | | |
| Mutoro | 1 | |
| Others | 1.13 (0.47–2.67) | 0.789 |
| Distance from home to facility | | |
| ≤5 km | 1 | |
| >5 km | 0.4 (0.34–0.89) | **0.006**** |
| Age of caretaker | | |
| <35 years | 1 | |
| 35–44 years | 0.93 (0.37–2.35) | 0.875 |
| >44 years | 0.70 (0.23–2.11) | 0.529 |
| HPV knowledge of caretaker | | |
| Low | 1 | |
| Moderate | 2.99 (1.14 – 7.87) | **0.026**** |
| High | 2.21 (0.86 – 5.69 | 0.102 |
| Impact of Religion on child HPV vaccination | | |
| Yes | 1 | |
| No | 0.61 (0.18–2.02) | 0.418 |
| Impact of clan on child HPV vaccination | | |
| Yes | 1 | |
| No | 0.57 (0.17–1.95) | 0.374 |

*****Statistical significant ($p \le 0.05$) at multivariate analysis.*

The study by Migoneetal, demonstrates that inequity in uptake may persist in school based programmes while the difference in mean and median uptake between disadvantaged schools and other Irish schools in Irish study was small, the majority of schools with the lowest uptake (≤ 50%) were disadvantaged. Disadvantaged schools were twice as likely to have an uptake of ≤ 50% when compared with other schools, independent of other school characteristics [26].

Lower HPV vaccine uptake in disadvantaged schools has been reported by other studies; In Manchester, uptake was significantly lower in more deprived areas while [27].

Findings from a longitudinal study carried out in Eldoret, Kenya, where HPV vaccine acceptability was measured before a vaccination program (n = 287) and vaccine uptake, as reported by mothers, once the program was finished (n = 256) indicates that even though baseline acceptance was very high (88.1%), only 31.1% of the women reported at follow-up that their daughters had been vaccinated. The vaccine was declined by 17.7%, while another 51.2% had wanted the vaccination but were obstructed by practical barriers including cultural acceptance related barriers [16,28].

A study conducted in Lira district, Uganda, on the level and factors associated with the uptake of HPV vaccine among adolescent girls aged between 12 and 17 years also demonstrated low uptake where 49.6%(228/460) had not taken any of the vaccines, 18% (83/460) had received one dose, 14.8%(68/460) had received two doses and the uptake was associated with factors like education and other social economic factors [17,29].

Knowledge of HPV vaccination and distance to facility were the factors associated with uptake of Human papilloma virus vaccine among adolescent girls aged 11 to 15 years in Kabarole district.

Baseline information on knowledge, attitude and practice towards HPV vaccination was crucial in establishing a progress track on the current HPV immunization program [30]. In a study conducted in Malaysia with a total sample of 380 respondents who participated in this study. Females scored significantly higher for the knowledge items compared to the males. Majority of respondents (86.6%) indicated their intention to get HPV vaccines. Willingness to be vaccinated was significantly associated with the level of knowledge of cervical cancer (AOR 1.66; 95% CI 1.018–2.698; p = 0.042). Gender (AOR 3.29; 95% CI 2.00–5.41; p < 0.001) lack of knowledge was found to be a significant predictor for someone who rejects vaccination due to side effects [30].

The study conducted by Satterwhite CL in Malaysia concluded that knowledge of HPV and its preventive measures among the respondents were still insufficient. Attitude towards HPV vaccination was significantly associated with knowledge about cervical cancer [8]. In addition, vaccination practice among secondary school girls was high, indicating that the national HPV immunization program was effective in delivering the HPV vaccine [10].

Since acceptance of HPV vaccination varies internationally, and many adolescents were still not getting the HPV vaccine in various countries [31], it was important to understand why some parents choose to vaccinate their children and some parents do not in order to continue to increase vaccination uptake. According to the study conducted by Brooke Nickel 2017, both low and high HPV knowledge may be associated with lower rates of vaccination, with parents' country and gender also being influential factors. It also demonstrated that parental attitudes towards the HPV vaccine differ by country and knowledge [32].

Given that the primary target population for HPV vaccination program was girls aged 9–13, typically before the initiation of sexual accident, parental knowledge and attitudes play an important role in the success of vaccination as consent was usually required for their adolescent children to be vaccinated. Research aimed at understanding HPV vaccine uptake has demonstrated that uptake of the HPV vaccine was generally high with good knowledge about the vaccine; however parents and girls often had insufficient knowledge and understanding about and had varying attitudes towards vaccination [13]. Alongside this, several studies conducted across different settings have aimed to examine factors influencing HPV vaccine uptake. Findings from these studies were wide-ranging, however, parental intentions have been shown to consistently be a strong predictor of their children's HPV vaccine uptake [20].

According to the findings by Brooke 2017, the strongest factor associated with daughters' vaccination status across the entire sample was parents' HPV knowledge ($p < 0.001$). Parents' HPV knowledge scores displayed a non-linear relationship; parents with low knowledge scores and parents with high knowledge scores were less likely to have vaccinated their daughters. HPV vaccination specific knowledge was also significant univariate factor associated with vaccination status ($p < 0.05$) [13].

Vaccination specific knowledge and very high levels of vaccination specific knowledge were also less likely to have vaccinated their daughters. Parents' demographic characteristics including their country of origin (OR = 2.2, 95% CI: 1.07–4.50; $p < 0.05$) and gender (OR = 0.5 95% CI: 0.26–0.94, $p < 0.05$) were also factors associated with non-vaccination, with parents in the US and men (across all countries) being less likely to vaccinate their daughters [26].

This is also in line with a study conducted in Ethiopia Debre Tabar Town by Gedefaye in which secondary education and above (AOR 1.70, 95% CI 1.05–2.27) and having good knowledge of the HPV vaccine (AOR 3.30, 95% CI 2.21–4.93) were significantly associated with willingness to receive the HPV vaccine.

From the qualitative analysis, it was reported that some villages in Kabende are hard to reach and it is worse in the rainy season when the road network is poor. This is consistent with the findings on the quantitative analysis where long distance to the health facilities [27,33,34] was associated with poor uptake of the vaccination. The long distance with a poor road network makes it hard for the adolescents to access the vaccination points [26].

In Summary, the findings in this study are consistent with the results in other studies where the uptake and completion of the vaccination programme is still a challenge [16,35].

Also the findings on the factors associated with the uptake from this study are consistent with the findings in other studies where knowledge about the vaccine by both the adolescent girl and caretaker, level of education by the caretaker and accessibility to the vaccination points are strongly associated with uptake of the vaccine [1,36–38].

From this study, it was also established that most HPV vaccination programmes target girls aged less than 12 years old or those in primary school, which enables them to get the best protection. Most girls at secondary school level have already started sexual debut, so it is very important for them to complete the HPV vaccination before they start secondary education [39–42].

## Conclusion

In conclusion, this study provides a preliminary insight into the uptake of human papilloma virus vaccine among adolescent girls in Kabarole district. HPV vaccination uptake was at 63% and relatively lower than national average. Education of caretaker, Knowledge of HPV vaccination and distance to facility were the factors associated with uptake of Human papilloma virus vaccine among adolescent girls aged 11 to 15 years in Kabarole district.

### Ethics approval and consent to participate

**Norm/Standard according to which research was conducted .**  This research was conducted adhering to guidelines set forth by the Declaration of Helsinki, which outlines ethical principles for medical research involving human subjects. The research was approved by the Research and Ethics committee of the Faculty of Medicine of Mbarara University of science and technology, and to the research and ethics committee of Mbarara University of science and technology.

The study protocol was aligned with ethical guidelines. Informed consent was obtained from study participants and their guardians, Participants in the study were ensured of confidentiality of their data, minimization of potential harm and discomfort, and respect for participant autonomy and dignity was maintained.

Additionally, the study adhered to relevant national and institutional regulations regarding research involving human subjects and disclosed any potential conflicts of interest. Authorization to carry out the study was obtained from the health sub district authorities, education officer, and the district health office authorities in the study area. More so authorization to interview health workers was obtained from their facility managers before they could participate in the study. The objectives of the study were explained to study participants and/or their guardians and written consent was obtained when consented to participate in the study. The consent form was signed in duplicate, and one copy remained with the participant. To ensure privacy and comfort to the participants, participant were interviewed from a selected private place at their home, workplace or school. All participants in the study were assured that they were free to pull out of the study in case they feel uncomfortable and that no consequences at all were to be gotten for the respondents who decide to move out of the study. The respondents were told that no names were required as the information given was solely for academic research purposes.

## Acknowledgements

The Authors thank the research assistants for their integrity and hard work in collecting the data, the teachers of the participating schools for their cooperation and the health workers who provided valuable information for this study. The authors also acknowledge all researchers whose findings helped in the development of this manuscript. The Authors also thank all the study participants- adolescent girls in Kabarole district, for accepting to participate in the study.

## Author contributions

**Conceptualization:** Tony Mugisa, Solomon Asiimwe, Fred N. Bagenda.

**Data curation:** Tony Mugisa.

**Formal analysis:** Tony Mugisa, Solomon Asiimwe.

**Investigation:** Tony Mugisa, Solomon Asiimwe.

**Methodology:** Solomon Asiimwe.

**Resources:** Solomon Asiimwe.

**Software:** Tony Mugisa.

**Supervision:** Fred N. Bagenda.

**Visualization:** Fred N. Bagenda.

**Writing – original draft:** Tony Mugisa, Solomon Asiimwe.

**Writing – review & editing:** Tony Mugisa, Fred N. Bagenda.

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
