## [Decision Letter · Decision Letter 0]

15 Sep 2024

PONE-D-24-25387Factors Associated with Uptake of Human Papilloma virus Vaccine among Adolescent girls: A cross sectional survey on insights into HPV Infection Prevention in Kabarole District -Western UgandaPLOS ONE

Dear Dr. Mugisa,

Thank you for submitting your manuscript to PLOS ONE. After careful consideration, we feel that it has merit but does not fully meet PLOS ONE’s publication criteria as it currently stands. Therefore, we invite you to submit a revised version of the manuscript that addresses the points raised during the review process. Address the specific comments related to study design, data analysis, interpretation of statistical outputs, and inconsistency in the results and discussions.

We look forward to receiving your revised manuscript.

Kind regards,

Jonah Musa, MBBS, MSCI,PhD

Academic Editor

PLOS ONE

Journal Requirements:

Reviewers' comments:

Reviewer's Responses to Questions

**Comments to the Author**

1. Is the manuscript technically sound, and do the data support the conclusions?

Reviewer #1: Partly

Reviewer #2: Partly

2. Has the statistical analysis been performed appropriately and rigorously? 

Reviewer #1: N/A

Reviewer #2: No

3. Have the authors made all data underlying the findings in their manuscript fully available?

Reviewer #1: No

Reviewer #2: Yes

4. Is the manuscript presented in an intelligible fashion and written in standard English?

Reviewer #1: Yes

Reviewer #2: No

5. Review Comments to the Author

Reviewer #1: Very Respected Authors,

After carefully reading your paper I have few suggestions. Keywords are usually listed below the abstract. The objective of the study is clear. It is necessary to specify the methodology. Between December 18/10/2021 to 18/10/2022 we conducted a A cross-sectional study design line 21. I do not understand. I see that the study was done in October. Could you please, explain the sample size and the exact timing of when the study was conducted? The date of the Ethics Committee's decision and its reference number are required lines 103-105.

Reviewer #2: The research talks about a very important and topical issue especially in the aspect of elimination of cervical cancer.

But I have some major concerns with the manuscript

1. Data analysis: The values of the AOR 4.06 does not fall within the CI of (0.69-0.087), also OR of 0.4 does not fall within the CI of (1.43-8.09) all this are written under the abstract.

2. Under the results section: Even though the AOR and OR seem to fall within the CI but now it makes the whole results inconsistent. Kindly address that, because this ultimately affects interpretation and discussion of results.

3. The references are not serially numbered example under introduction from number 1, the next reference was (42), and it also seem to be inconsistent having a mix of both APA and Vancouver styles. Please address that.

4. Data collection: what's the rationale of using 10 for pretesting.

4. Other minor issues include the typographical errors.

6. PLOS authors have the option to publish the peer review history of their article (what does this mean? ). If published, this will include your full peer review and any attached files.

**Do you want your identity to be public for this peer review?** For information about this choice, including consent withdrawal, please see our Privacy Policy .

Reviewer #1: No

Reviewer #2: **Yes: ** Maryam Jamila Ali

---

## [Author Response · Author response to Decision Letter 1]

24 Oct 2024

Dear Reviewer,,

Response to reviewers on the Manuscript “ Factors Associated with Uptake of Human Papilloma virus Vaccine among Adolescent girls: A cross sectional survey on insights into HPV Infection Prevention in Kabarole District -Western Uganda”

Reference is made to the comments by email on the required changes to make in the manuscript “ Factors Associated with Uptake of Human Papilloma virus Vaccine among Adolescent girls: A cross sectional survey on insights into HPV Infection Prevention in Kabarole District -Western Uganda”

The comments have been addressed by the author as guided and a re-submission is made to that effect.

I appreciate the very educative guidance provided in the comments

Details and pages of the responses

Comment Response

1.Please ensure that your manuscript meets PLOS ONE's style requirements, including those for file naming. The manuscript has been checked against the PLOS ONE guide for authors and now meets the requirements

2. We note that your Data Availability Statement is currently as follows: [All relevant data are within the manuscript and its Supporting Information files]. Please confirm at this time whether or not your submission contains all raw data required to replicate the results of your study. I confirm that all all relevant data are with in the manuscript and its supporting information files

3. Please include your full ethics statement in the ‘Methods’ section of your manuscript file. In your statement, please include the full name of the IRB or ethics committee who approved or waived your study, as well as whether or not you obtained informed written or verbal consent. If consent was waived for your study, please include this information in your statement as well. A complete Ethics statement has been included in the Methods section and included the Full name of the ethics committee who approved the study as well as the statement on how consent was obtained.

Page 10 - 11

4. Reviewer #1: Very Respected Authors,After carefully reading your paper I have few suggestions. Keywords are usually listed below the abstract. Correction has been made, Key words have been transferred to below the abstract.

Page 2

5. The objective of the study is clear. It is necessary to specify the methodology. Between December 18/10/2021 to 18/10/2022 we conducted a A cross-sectional study design line 21. I do not understand. I see that the study was done in October. Could you please, explain the sample size and the exact timing of when the study was conducted? The date of the Ethics Committee's decision and its reference number are required lines 103-105.

The Methodology of the study have been revised. It clarifies that the study was conducted between December 18/10/2021 to 18/10/2022 we conducted a study in western Uganda. Actual data collection was done from 01/09/2022 to 18/10/2022. I have included on all the steps taken to arrive at the sample size. I have also included the Date and number of the Ethical committees decision [16/10/2021 under REC decision Number MUST-2021-73].

Page 10 &11

6.Reviewer #2: The research talks about a very important and topical issue especially in the aspect of elimination of cervical cancer.

But I have some major concerns with the manuscript I have addressed all the concerns suggested by the Reviewer

1.Data analysis: The values of the AOR 4.06 does not fall within the CI of (0.69-0.087), also OR of 0.4 does not fall within the CI of (1.43-8.09) all this are written under the abstract.

The value of AOR 4.06 has been corrected from the data and now falls with in the CI [(AOR=4.06; 95%CI (1.69 - 6.87); p=0.004)]. the OR of 0.4 has also been corrected and now show (OR=0.4; 95%CI (0.34 – 0.89) ; p=0.006)

Page 1, 2 and 11

2.Under the results section: Even though the AOR and OR seem to fall within the CI but now it makes the whole results inconsistent. Kindly address that, because this ultimately affects interpretation and discussion of results. The AOR and OR in the abstract and the results have been corrected and how falls in line. These were addressed by re-analyzing the data

Page 1, 2 and 11

3.The references are not serially numbered example under introduction from number 1, the next reference was (42), and it also seem to be inconsistent having a mix of both APA and Vancouver styles. Please address that. The references have been addressed and are all numbered serially and only in Vancouver style

4.Data collection: what's the rationale of using 10 for pretesting.

To achieve validity and reliability, consideration was made for diversity of respondents, number of questions in the tool, and the nature of the questions. A 5-10 people are acceptable. We went for the upper end of 10 people

5.Other minor issues include the typographical errors. Typographic errors have been corrected

Hoping for your consideration for approval of the manuscript,

I remain Yours Faithfully

Mr. Mugisa Tony

Assistant Lecturer- Public Health Department - Mountains of the Moon University

Corresponding Author

---

## [Editor Report · Decision Letter 1]

31 Oct 2024

Factors Associated with Uptake of Human Papilloma virus Vaccine among Adolescent girls: A cross sectional survey on insights into HPV Infection Prevention in Kabarole District -Western Uganda

PONE-D-24-25387R1

Dear Dr. Mugisa,

We’re pleased to inform you that your manuscript has been judged scientifically suitable for publication and will be formally accepted for publication once it meets all outstanding technical requirements.

Kind regards,

Jonah Musa, MBBS, MSCI,PhD

Academic Editor

PLOS ONE
---

## [Editor Report · Acceptance letter]

PONE-D-24-25387R1

PLOS ONE

Dear Dr. Mugisa,

I'm pleased to inform you that your manuscript has been deemed suitable for publication in PLOS ONE. Congratulations! Your manuscript is now being handed over to our production team.

Kind regards,

on behalf of

Dr. Jonah Musa

Academic Editor

PLOS ONE